

SciPost Phys. Lect. Notes 3 (2018)

# Birdtracks for $\mathrm{SU}(N)$

**Stefan Keppeler**

Fachbereich Mathematik, Universität Tübingen,
Auf der Morgenstelle 10, 72076 Tübingen, Germany

★ stefan.keppeler@uni-tuebingen.de

## Abstract

**I gently introduce the diagrammatic birdtrack notation, first for vector algebra and then for permutations. After moving on to general tensors I review some recent results on Hermitian Young operators, gluon projectors, and multiplet bases for $\mathrm{SU}(N)$ color space.**

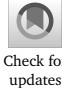
## Introduction

The term *birdtracks* was coined by Predrag Cvitanović, figuratively denoting the diagrammatic notation he uses in his book on Lie groups [1] – hereinafter referred to as THE BOOK. The birdtrack notation is closely related to (abstract) index notation. Translating back and forth between birdtracks and index notation is achieved easily by following some simple rules. Birdtracks, however, avoid the otherwise frequent cluttering of indices in longer expressions. Being a notation for all sorts of tensors, besides applications in representation theory, birdtracks are useful in a wide range of topics, from differential geometry and general relativity (see, e.g., [2]) to the classification of semisimple Lie algebras (*exceptional magic* in THE BOOK). More details (and references) can, e.g., be found in Sec. 4.9 of THE BOOK.

These notes were prepared for an 8 hour course aimed at graduate students at the QCD MASTER CLASS 2017 held from 18 to 24 June 2017 in Saint-Jacut-de-la-Mer, France. I introduce the birdtrack notation twice in well-known areas, first for vector algebra in Sec. 1 and later for permutations in Sec. 3. The notation is extended and adapted to SU($N$) tensors in Sec. 2. In Sec. 4 I demonstrate the usefulness of birdtracks for calculations in quantum chromodynamics (QCD). In particular, I discuss the construction of bases for QCD colour space. The intention of these notes is thus twofold: Sections. 1-3 contain a gentle introduction to the birdtack notation, whereas Sec. 4 illustrates the use of birdtracks for QCD colour structure.

If you would like to learn more about different flavours of birdtracks, their history and their uses, then jump to Sec. 5. First, however, I'd rather you joined me on a little journey into the world of birdtracks, starting with basic vector algebra in the following section.

## 1  Vector algebra

We begin with vectors $\vec{a}, \vec{b}, \vec{c}, \ldots \in \mathbb{R}^3$, scalar and cross products, review the index notation and introduce the diagrammatic birdtrack notation as illustrated in the following table.

| | index notation | birdtrack notation |
|---|---|---|
| vector $\vec{a}$ | $a_j$ | |
| scalar product $\vec{a} \cdot \vec{b}$ | $a_j b_j$ | |
| cross product $\vec{a} \times \vec{b}$ | $\varepsilon_{jk\ell} a_j b_k$ | |

More precisely, translating back and forth between index notation and birdtracks is achieved by assigning indices to external lines,

$$a_j = \quad \boxed{a} \!\!-\!\!-^{j} \quad . \tag{1}$$

Index contractions (we always sum over repeated indices) correspond to joining lines,

$$a_j b_j = \quad \boxed{a}\!\!-\!\!-\!\!\boxed{b} \quad . \tag{2}$$

Consequently, an isolated line is a Kronecker-$\delta$,

$$\delta_{jk} = \quad {}^{j}\underline{\qquad}^{k} \quad , \tag{3}$$

(contracting with a Kronecker-$\delta$ corresponds to extending a line). For the totally anti-symmetric $\varepsilon$,

$$\begin{aligned}
\varepsilon_{123} &= \varepsilon_{231} = \varepsilon_{312} = 1 \\
\varepsilon_{213} &= \varepsilon_{132} = \varepsilon_{321} = -1 \\
\varepsilon_{jk\ell} &= 0 \quad \text{if at least two indices have the same value,}
\end{aligned} \tag{4}$$

we write a vertex,

$$\varepsilon_{jk\ell} = \quad \tag{5}$$

thereby agreeing to read off indices in counter-clockwise order.

When not assigning indices – which is what we want to do most of the time – the position where an external line ends determines which lines have to be identified in equations. For instance, the anti-symmetry of $\varepsilon$ in the first two indices is expressed as

$$\tag{6}$$

Now we can use this notation to write components of cross products,

$$\varepsilon_{jk\ell} a_j b_k = \quad \tag{7}$$

where we omit labelling lines with indices over which we sum anyway. Equivalently, omitting indices altogether,

$$\vec{a} \times \vec{b} = \quad \tag{8}$$

Let's study the following diagram,

$$\tag{9}$$

Assigning indices for a moment,

$$\tag{10}$$

we see that the only non-zero terms have

$$\begin{aligned}
i &= k \text{ and } j = \ell, \text{ or} \\
i &= \ell \text{ and } j = k,
\end{aligned}$$

since otherwise we cannot satisfy $i \neq j \neq m \neq i$ and $m \neq \ell \neq k \neq m$. Thus, (9) is a linear combination of the diagrams

$$\text{and} \qquad . \tag{11}$$

Closer inspection shows

$$= \quad - \quad , \tag{12}$$

which is nothing but the birdtrack version of the well-known identity

$$\varepsilon_{ijm}\varepsilon_{\ell km} = \delta_{i\ell}\delta_{jk} - \delta_{ik}\delta_{j\ell} . \tag{13}$$

(Of course, we could have also obtained Eq. (12) by translating Eq. (13) into birdtrack notation instead of deriving the identity within birdtrack notation.)

Equation (12) can be used in order to derive identities for double cross products and similar formulas from vector algebra which are notoriously difficult to remember. For instance,

$$(\vec{a} \times \vec{b}) \times \vec{c} =$$

$$= \quad - \quad = (\vec{a} \cdot \vec{c})\vec{b} - (\vec{b} \cdot \vec{c})\vec{a} . \tag{14}$$

**Exercise 1** *Derive a similar identity for* $(\vec{a} \times \vec{b}) \cdot (\vec{c} \times \vec{d})$.

**Exercise 2** *Show that* $\big((\vec{a} \times \vec{b}) \times \vec{c}\big) \times \vec{d} = \big((\vec{a} \times \vec{b}) \cdot \vec{d}\big)\vec{c} - (\vec{a} \times \vec{b})(\vec{c} \cdot \vec{d})$.

In birdtrack notation, it is immediately manifest that the triple product,

$$(\vec{a} \times \vec{b}) \cdot \vec{c} = \quad , \tag{15}$$

is invariant under cyclic permutations of the three vectors.

Taking Eq. (12) and joining the upper left to the upper right line and also the lower left to the lower right line yields

$$= \quad - \quad = 3 - 3 \cdot 3 = -6, \tag{16}$$

where we have used that each loop contributes a factor of $\delta_{jj} = 3$.

**Exercise 3** *Evaluate*

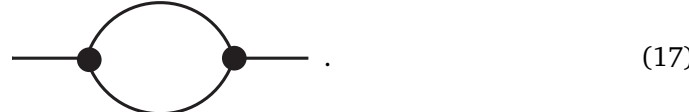

$$\tag{17}$$

Rotating Eq. (12) by 90° we obtain the equivalent identity

$$\tag{18}$$

**Exercise 4** *Evaluate*

$$\tag{19}$$

Finally, we want to study

$$\tag{20}$$

Imagine for a moment assigning indices to the lines, then the diagram is non-zero only if the value of each index on the left matches the value of exactly one index on the right, i.e.

$$= A \quad + B \quad + C$$
$$+ D \quad + E \quad + F \, . \tag{21}$$

with some constants $A$ to $E$. Intertwining, say the first two lines on the right, introduces a factor of $(-1)$ on the l.h.s. of the equation, see eq. (6), whereas on the r.h.s. the roles of the terms interchange. Together this implies

$$B = -A, \quad F = -C \quad \text{and} \quad E = -D \, . \tag{22}$$

Similarly, by intertwining the lower two lines, we find

$$C = -A, \quad E = -B \quad \text{and} \quad F = -D \, , \tag{23}$$

and thus

$$= A \left( \quad - \quad - \right.$$
$$\left. - \quad + \quad + \quad \right) . \tag{24}$$

The factor $A$ can, e.g., be determined by joining all lines on the left to those on the right (first to first etc.): On the l.h.s. we obtain $-6$, see Eq. (16), and hence

$$
\begin{aligned}
-6 = A &\left( \vphantom{\bigg(} \right. \cdots \\
&= A(27 - 9 - 9 - 9 + 3 + 3) \\
&= 6A \quad \Longleftrightarrow \quad A = -1.
\end{aligned}
\tag{25}
$$

Later, we will denote anti-symmetrisation of a couple of lines by a solid bar over these lines, normalised by the number of terms, i.e.

$$
= \frac{1}{3!} \left( \cdots \right),
\tag{26}
$$

and likewise for symmetrisation using an open bar. With this notation we can rewrite our result as

$$
= -6 \qquad .
\tag{27}
$$

## 2 Birdtracks for SU($N$) tensors

In Sec. 1 we have encountered a simple rule for translating expressions from index notation to birdtrack notation: Draw some vertex, box or blob, possibly with a name inside and attach a line for each index.

In later sections we will also study quantities that have several different types of indices, and therefore we will use different types of lines. In particular, we are interested in the following situation. Let $V$ be a finite dimensional (complex) vector space, say $\dim V = N$ (i.e. $V \cong \mathbb{C}^N$) and let $\overline{V}$ be its dual, i.e. the space of all linear maps $V \to \mathbb{C}$. In index notation we denote components[1] of $v \in V$ by $v^j$, with an upper index $j = 1, \ldots, N$. Components of $u \in \overline{V}$ are in turn denoted by $u_j$, with a lower index $j = 1, \ldots, N$. In order to distinguish the two kinds of indices we write

$$
v^j = \boxed{v} \!\!-\!\!\!\leftarrow^{\,j} \qquad \text{and} \qquad u_k = \boxed{u} \!\!\rightarrow^{\,k} \,,
\tag{28}
$$

i.e., arrows point away from upper indices and towards lower indices. Since we can only contract upper indices with lower indices, in birdtrack notation we can only connect lines whose arrows point in the same direction, e.g.

$$
u(v) = u_j v^j = \boxed{u} \!\!\rightarrow\!\!\boxed{v} \,,
\tag{29}
$$

_______________

[1] in abstract index notation also the vector itself

and, consequently, $\delta^j_{\;k} = {}_{j}\!\!\longrightarrow\!\!{}_{k}$ .

If $V$ carries a representation $\Gamma$ of a Lie group $G$, i.e. $\Gamma : G \to \mathrm{GL}(V)$, then $\overline{V}$ naturally carries the contragredient representation, for which the representation matrices are the transposes of the inverses. If the representation $\Gamma$ is unitary, then the inverse transpose is the complex conjugate.

We are particularly interested in the case where $V = \mathbb{C}^N$ and where $\Gamma$ is the defining (or fundamental) representation of $G = \mathrm{SU}(N)$. Then $\overline{V}$ carries the complex conjugate of the defining representation. Now complex conjugation of a diagram is achieved by reversing all arrows.

Another important representation is the adjoint representation, the representation of a Lie group $G$ on its own Lie algebra $\mathfrak{g}$. For $G = \mathrm{SU}(N)$, the Lie algebra $\mathfrak{g} = \mathrm{su}(N)$ consists of the traceless Hermitian $N \times N$ matrices. Elements of a basis of the Lie algebra are called generators, and $\mathrm{su}(N)$ is a real vector space of dimension $N^2 - 1$. Since it is a real vector space we do not distinguish upper and lower indices, and in birdtrack notation we introduce a new type of line without arrow.[2] In particular, we denote generators $t^a \in \mathrm{su}(N)$ as

$$(t^a)^j_{\;k} = \quad {}_{j}\!\!\longrightarrow\!\!{}_{k}^{\;a} \quad , \tag{30}$$

i.e. we write no vertex symbol, box or blob.

Having applications in quantum chromodynamics (QCD) in mind, we also refer to upper and lower indices $j, k, \ell \in \{1, \ldots, N\}$ as quark and anti-quark indices, to lines with arrows as (anti-)quark lines, to indices $a, b, c \in \{1, \ldots, N^2 - 1\}$ as gluon indices, and to curly lines as gluon lines.

Cvitanović [1] draws gluon lines as thin straight lines instead of curly lines, and in handwritten notes it is often convenient to use wiggly lines.

Vanishing of the trace of the generators, $\mathrm{tr}\, t^a = (t^a)^j_{\;j} = 0$, is expressed as

$$\text{\large(gluon loop)} = 0 \tag{31}$$

in birdtrack notation.

The Lie bracket (commutator) is given by

$$[t^a, t^b] = t^a t^b - t^b t^a = \mathrm{i} f^{abc} t^c , \tag{32}$$

with the totally anti-symmetric structure constants $f^{abc}$. The latter we denote by a vertex,

$$\mathrm{i} f^{abc} = \quad \text{\large(vertex diagram)} \quad , \tag{33}$$

reading off indices in anti-clockwise order.

Written in components Eq. (32) reads

$$(t^a)^j_{\;\ell}(t^b)^\ell_{\;k} - (t^b)^j_{\;\ell}(t^a)^\ell_{\;k} = \mathrm{i} f^{abc}(t^c)^j_{\;k} , \tag{34}$$

---

[2]However, in order to be able to take tensor products of $V$, $\overline{V}$ and the carrier space of the adjoint representation we complexify $\mathrm{su}(N)$ to $A \cong \mathbb{C}^{N^2-1}$.

whereas in birdtrack notation we write

$$\text{(35)}$$

It is convenient to normalise the generators as follows,

$$\mathrm{tr}(t^a t^b) = T_R\,\delta^{ab} \quad\Longleftrightarrow\quad (t^a)^j{}_k (t^b)^k{}_j = T_R\,\delta^{ab}$$

**matrix notation**            **index notation**

$$\Longleftrightarrow \qquad = T_R \qquad , \tag{36}$$

**birdtrack notation**

where $T_R$ is an arbitrary normalisation constant; $T_R = \frac{1}{2}$, 1 or 2 are common choices. Multiplying the Lie bracket (35) with another generator and taking the trace we obtain

$$\text{(37)}$$

which can be rewritten as

$$= T_R \tag{38}$$

**Exercise 5** *Re-derive Eq. (38) in matrix or index notation.*

**Exercise 6** *Decomposition of $\overline{V} \otimes V$.*
   *We define the linear map $P_A : \overline{V} \otimes V \to \overline{V} \otimes V$ by*

$$(P_A)^j{}_k{}^\ell{}_m = \frac{1}{T_R}\,(t^a)^j{}_k (t^a)^\ell{}_m\,. \tag{39}$$

**a)**    *Write $P_A$ in birdtrack notation and verify that $(P_A)^2 = P_A$ (in birdtracks!).*

*We further define the linear map $P_\bullet : \overline{V} \otimes V \to \overline{V} \otimes V$ by*

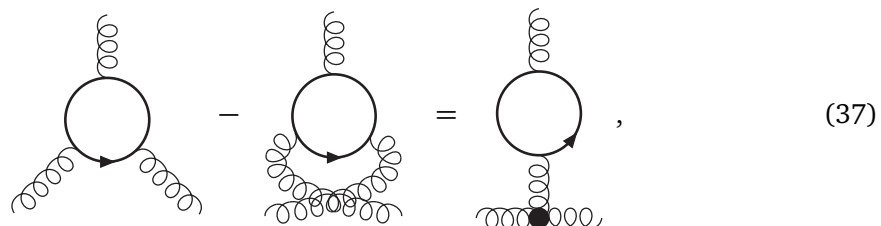

$$P_\bullet = C \qquad . \tag{40}$$

**b)**    *Fix $C > 0$ such that $P_\bullet^2 = P_\bullet$.*

*Now $P_\bullet$ and $P_A$ are projection operators. We find the dimensions of their images (the subspaces onto which they project) by taking the trace.*

**c)** *Determine* $\operatorname{tr} P_\bullet$ *and* $\operatorname{tr} P_A$ *(in birdtracks!).*

**d)** *Calculate* $P_\bullet P_A$ *and* $P_A P_\bullet$ *(in birdtracks!).*

*Apparently,* $P_\bullet$ *and* $P_A$ *project onto mutually transversal[3] subspaces. From* $\operatorname{tr} P_\bullet + \operatorname{tr} P_A = N^2$ *we conclude that* $P_\bullet + P_A = \mathbb{1}_{\overline{V} \otimes V}$, *in birdtracks*

$$\longrightarrow\!\!\!\!\!\!\longleftarrow = \frac{1}{N} \;\big)\!\big( \; + \; \frac{1}{T_R} \;\big)\!\!\sim\!\!\big( \;. \tag{41}$$

Equation (41) can be rearranged to yield the important identity (sometimes referred to as Fierz identity)

$$\big)\!\!\sim\!\!\big( = T_R \longrightarrow\!\!\!\!\!\!\longleftarrow - \frac{T_R}{N} \;\big)\!\big( \;, \tag{42}$$

which can be used in order to remove internal gluon lines from any diagram. For instance,

$$\longrightarrow = T_R \longrightarrow - \frac{T_R}{N} \longrightarrow$$
$$= T_R \frac{N^2-1}{N} \longrightarrow . \tag{43}$$

The prefactor $C_F := T_R \frac{N^2-1}{N}$ is known as the (quadratic) Casimir operator in the defining/fundamental representation.

**Exercise 7** *Evaluate*

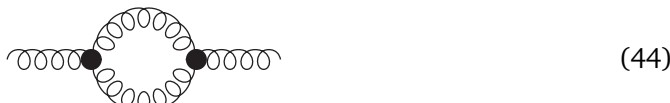

$$\tag{44}$$

*by first replacing the structure constants with quark loops according to Eq. (38) and then removing all internal gluon lines by means of Eq. (42). (You should obtain* $2T_R N \;\sim$ ; *the prefactor* $C_A := 2T_R N$ *is known as the (quadratic) Casimir operator in the adjoint representation.)*

**Exercise 8** *We would expect Eq. (35) to also hold with the quark line* $\longrightarrow$ *replaced by a gluon line* $\sim$ *(and the quark-gluon vertices/generators* $\genfrac{}{}{0pt}{}{}{}$ *replaced by triple-gluon vertices/structure constants* $\sim\!\!\!\bullet\!\!\!\sim$ *) – the resulting equation being known as Jacobi identity. Prove the Jacobi identity in birdtracks as follows:*

**a)** *Replace the vertices in* $\genfrac{}{}{0pt}{}{\sim\!\!\bullet\!\!\sim}{\sim\!\!\bullet\!\!\sim}$ *by quark loops according to Eq. (38). Then remove all internal gluon lines by using Eq. (42).*

**b)** *Rotate the result of (a) by 90°.*

**c)** *Cross the two lower lines of your result of (b).*

---

[3]i.e. $\operatorname{im} P_\bullet \subset \ker P_A$ and $\operatorname{im} P_A \subset \ker P_\bullet$.

**d)** *Subtract your result of (c) from your result of (b) and compare with (a).*
*This should conclude the proof of the Jacobi relation.*

The transpose of a birdtrack diagram is obtained by mirroring the diagram across a vertical line, e.g.

$$(P_\bullet)^T = C \quad \rangle \quad \langle \quad . \tag{45}$$

Hermitian conjugation, denoted by a dagger, corresponds to transposing and taking the complex conjugate; recall that complex conjugation amounts to reversing all arrows.

**Exercise 9** *Verify that $(P_A)^\dagger = P_A$ and $(P_\bullet)^\dagger = P_\bullet$, with $P_A$ and $P_\bullet$ as defined in Eqs. (39) and (40).*

Recall that Hermitian projection operators project orthogonally. Let us illustrate this statement for projections from the $xy$-plane to the coordinate axes. We can, e.g., project a point $\begin{pmatrix} x \\ y \end{pmatrix}$ to the $x$-axis or to the $y$-axis by applying the operators

$$P_x = \begin{pmatrix} 1 & 0 \\ 0 & 0 \end{pmatrix} \quad \text{or} \quad P_y = \begin{pmatrix} 0 & 0 \\ 0 & 1 \end{pmatrix}, \tag{46}$$

respectively. $P_x$ and $P_y$ are projectors, since $P_x^2 = P_x$ and $P_y^2 = P_y$, their images being the $x$- and the $y$-axis, respectively.
However, we can also project to the coordinate axes in many other ways. For instance,

$$Q_x = \begin{pmatrix} 1 & 1 \\ 0 & 0 \end{pmatrix} \quad \text{or} \quad Q_y = \begin{pmatrix} 0 & 0 \\ 1 & 1 \end{pmatrix}, \tag{47}$$

also satisfy and $Q_x^2 = Q_x$ and $Q_y^2 = Q_y$, as well as

$$
\begin{aligned}
\operatorname{im} Q_x &= \operatorname{im} P_x = \{(x, y) \in \mathbb{R}^2 \mid y = 0\} \quad \text{and} \\
\operatorname{im} Q_y &= \operatorname{im} P_y = \{(x, y) \in \mathbb{R}^2 \mid x = 0\}.
\end{aligned} \tag{48}
$$

A projection operator projects onto its image along (lines parallel to) its kernel, see figure. The kernels are

$$
\begin{aligned}
\ker P_x &= \{(x, y) \in \mathbb{R}^2 \mid x = 0\}, \\
\ker P_y &= \{(x, y) \in \mathbb{R}^2 \mid y = 0\}, \\
\ker Q_x &= \{(x, y) \in \mathbb{R}^2 \mid y = -x\} = \ker Q_y.
\end{aligned} \tag{49}
$$

If a linear operator is Hermitian then its image is orthogonal to its kernel. We have $P_x^\dagger = P_x$ and $P_y^\dagger = P_y$, and indeed $\operatorname{im} P_x \perp \ker P_x$ and $\operatorname{im} P_y \perp \ker P_y$. Moreover, since $\operatorname{im} P_x$ and $\operatorname{im} P_y$ intersect only at the origin, we have $P_x P_y = 0 = P_y P_x$. However, for the non-Hermitian projectors $Q_x$ and $Q_y$ we have $Q_x Q_y \neq 0 \neq Q_y Q_x$, although their images also intersect only at the origin.

## 3 Permutations and the symmetric group

Inspecting once more Eqs. (21), (24) and (26) we can come up with a different meaning for the diagrams on the right-hand side. Apparently, we have just found a notation for permutations of 3 objects, i.e. elements of the symmetric group $S_3$. (We denote by $S_n$ the group of all permutations of $n$ objects, the group multiplication being composition of mappings.) To this end read the diagrams on the r.h.s. of Eq. (21), (24) or (26) as mapping ends of lines from right to left. Recalling two other standard notations for permutations we have, e.g.,

$$\pi = \underbrace{\begin{pmatrix} 1 & 2 & 3 \\ 3 & 1 & 2 \end{pmatrix}}_{\text{two-line notation}} = \underbrace{(132)}_{\text{cycle notation}} = \underbrace{\text{diagram}}_{\text{birdtrack notation}} \,, \tag{50}$$

which all mean $\pi(1) = 3$, $\pi(2) = 1$, and $\pi(3) = 2$. Composition with a second permutation, e.g.

$$\sigma = \begin{pmatrix} 1 & 2 & 3 \\ 2 & 1 & 3 \end{pmatrix} = (12) = \text{diagram} \,, \tag{51}$$

can be determined in several ways. Say, we are interested in $\pi \circ \sigma$, we can

- determine individual elements

$$
\begin{aligned}
(\pi \circ \sigma)(1) &= \pi(\sigma(1)) = \pi(2) = 1 \\
(\pi \circ \sigma)(2) &= \pi(\sigma(2)) = \pi(1) = 3 \\
(\pi \circ \sigma)(3) &= \pi(\sigma(3)) = \pi(3) = 2 \,,
\end{aligned}
\tag{52}
$$

- multiply cycles (recall that every permutation is a product of disjoint cycles)

$$\pi \circ \sigma = (132)(12) \underset{\text{omit ‘}\circ\text{’}}{=} \underset{(*)}{(1)(23)} = \underbrace{(23)}_{\text{omit one-cycles}} \tag{53}$$

  (∗) **Write ‘(1’, where is it mapped? Thereby read from right to left.**
  **Continue till you'd return to $1$, then ‘)’**
  **Repeat starting with first number not used so far.**

  or

- compose diagrams,

$$\pi \circ \sigma = \text{diagram} = \text{diagram} \,, \tag{54}$$

  and twist lines at will – it only matters where lines enter and leave.

Finally, we obtain

$$\pi \circ \sigma = \begin{pmatrix} 1 & 2 & 3 \\ 1 & 3 & 2 \end{pmatrix} = (23) = \text{diagram} \,, \tag{55}$$

**Exercise 10** *Determine $\sigma \circ \pi$ in three different ways.*

**Exercise 11** *Write*

$$\begin{pmatrix} 1 & 2 & 3 & 4 & 5 \\ 2 & 4 & 5 & 1 & 3 \end{pmatrix} \in S_5 \tag{56}$$

  *in cycle and birdtrack notation.*

**Exercise 12** *Digression: Speaking about cycle notation, watch the video "An Impossible Bet" by* `minutephysics`, *www.youtube.com/watch?v=eivGlBKlK6M, (but not the solution!) and come up with a good strategy.*

Viewing the individual diagrams on the r.h.s. of Eqs. (21), (24) and (26) as permutations, the total expression is not an element of the group $S_3$ but of the group algebra $\mathscr{A}(S_3)$. Recall that the group algebra $\mathscr{A}(G)$ of a finite group $G$ is the vector space spanned by formal linear combinations of the group elements, with a multiplication induced from the group multiplication.

We define symmetrisers $S$ and anti-symmetrisers $A$ by

$$S = \frac{1}{n!} \sum_{\pi \in S_n} \pi \qquad \text{and} \qquad A = \frac{1}{n!} \sum_{\pi \in S_n} \text{sign}(\pi)\, \pi\,, \tag{57}$$

and denote them by open and solid bars, respectively,

$$S = \quad\quad \text{and} \quad A = \quad. \tag{58}$$

For instance, see also Eq. (26),

$$\tag{59}$$

Notice that in birdtrack notation the sign of a permutation, $(-1)^K$, is determined by the number $K$ of line crossings; if more than two lines cross in a point, one should slightly perturb the diagram before counting, e.g. $\rightsquigarrow$ ($K{=}3$).

**Exercise 13** *Expand* ▮ *and* ▯ *as in Eq. (59).*

We use the corresponding notation for partial (anti-)symmetrisation over a subset of lines,

e.g.

$$\text{(diagram)} = \frac{1}{2}\left( \text{(diagram)} + \text{(diagram)} \right)$$

or

$$\text{(diagram)} = \frac{1}{2}\left( \text{(diagram)} - \text{(diagram)} \right)$$
$$= \frac{1}{2}\left( \text{(diagram)} - \text{(diagram)} \right). \tag{60}$$

The prefactor $1/n! = 1/|S_n|$ in Eq. (57) is chosen such that $S^2 = S$ and $A^2 = A$.

**Exercise 14** *Convince yourself that*

$$\left( \text{(diagram)} \right)^2 = \text{(diagram)} = \text{(diagram)} \quad and \quad A^2 = A. \tag{61}$$

It follows directly from the definition of $S$ and $A$ that when intertwining any two lines $S$ remains invariant and $A$ changes by a factor of $(-1)$, i.e.

$$\text{(diagram)} = \text{(diagram)} \quad and \quad \text{(diagram)} = - \text{(diagram)}. \tag{62}$$

This immediately implies that whenever two (or more) lines connect a symmetriser to an anti-symmetrizer the whole expression vanishes, e.g.

$$\text{(diagram)} = 0. \tag{63}$$

Symmetrisers and anti-symmetrisers can by built recursively. To this end notice that on the r.h.s. of

$$\text{(diagram)} = \frac{1}{n}\left( \text{(diagram)} + \text{(diagram)} + \ldots + \text{(diagram)} \right) \tag{64}$$

we have sorted the terms according to where the last line is mapped – to the $n$th, to the $(n-1)$th, ..., to the first line line. Multiplying with ▭ from the left and disentangling lines we obtain the compact relation

$$
\vdots\ \big|\ \vdots = \frac{1}{n}\left( \vdots\ \big|\ \vdots + (n-1)\ \vdots\ \big|\!\times\!\big|\ \vdots \right). \tag{65}
$$

Similarly for anti-symmetrisers:

$$
\vdots\ \big|\ \vdots = \frac{1}{n}\left( \vdots\ \big|\ \vdots - \times\!\big|\ \vdots + \ldots + (-1)^{n-1} \times\!\big|\ \vdots \right) \tag{66}
$$

$$
\vdots\ \big|\ \vdots = \frac{1}{n}\left( \vdots\ \big|\ \vdots - (n-1)\ \vdots\ \big|\!\times\!\big|\ \vdots \right).
$$

**Exercise 15** *Convince yourself that the signs in Eq.* (66) *are correct.*

## 3.1 Recap: group algebra and regular representation

The group algebra $\mathscr{A}(G)$ of a finite group $G$ (i.e. the $\mathbb{C}$-vector space spanned by formal linear combinations of the group elements, with multiplication induced from the group multiplication), carries the so-called regular representation of $G$. The regular representation can be completely reduced to a direct sum containing all irreducible representations of $G$. Irreducible invariant subspaces are obtained by right-multiplication with primitive idempotents $e_j \in \mathscr{A}(G)$. For $G = S_n$ we already know two such idempotents, the symmetriser $S$ and the anti-symmetriser $A$, see Eqs. (57) and (58). Primitive idempotents generating all irreducible representations of $S_n$ are the so-called Young operators.

## 3.2 Recap: Young operators

Young diagrams are arrangements of $n$ boxes in $r$ rows of non-increasing lengths. A Young tableau $\Theta$ is a Young diagram with each of the numbers $1, \ldots, n$ written into one of its boxes. For a so-called standard Young tableau the numbers increase within each row from left to right and within each column from top to bottom. We denote the set of all standard Young tableaux with $n$ boxes by $\mathscr{Y}_n$, e.g.

$$
\mathscr{Y}_2 = \left\{ \boxed{1\,2},\ \boxed{\begin{smallmatrix}1\\2\end{smallmatrix}} \right\}, \quad \mathscr{Y}_3 = \left\{ \boxed{1\,2\,3},\ \boxed{\begin{smallmatrix}1&2\\3\end{smallmatrix}},\ \boxed{\begin{smallmatrix}1&3\\2\end{smallmatrix}},\ \boxed{\begin{smallmatrix}1\\2\\3\end{smallmatrix}} \right\}. \tag{67}
$$

Removing the box containing the number $n$ from $\Theta \in \mathscr{Y}_n$ we obtain a standard tableau $\Theta' \in \mathscr{Y}_{n-1}$.

For $\Theta \in \mathscr{Y}_n$ let $\{h_\Theta\}$ be the set of all horizontal permutations, i.e. $h_\Theta \in S_n$ leaves the sets of numbers appearing in the same row of $\Theta$ invariant. Analogously, vertical permutations $v_\Theta$ leave the sets of numbers appearing in the same column of $\Theta$ invariant. The Young operator $Y_\Theta$ is then defined in terms of the row symmetrizer, $s_\Theta = \sum_{\{h_\Theta\}} h_\Theta$, and the column anti-symmetrizer, $a_\Theta = \sum_{\{v_\Theta\}} \text{sign}(v_\Theta) v_\Theta$, as

$$Y_\Theta = \tfrac{1}{|\Theta|} s_\Theta a_\Theta \,. \tag{68}$$

Note that as opposed to Eq. (57) we have not included normalising factorials. The normalisation factor is given by the product of hook lengths of the boxes of $\Theta$, and thus depends only the shape of the Young tableau, i.e. on the Young diagram. The hook length of a given box counts the number of boxes below and to the right of this box, adding one for the box itself. For illustration we write the hook lengths into the boxes of a couple of Young diagrams,

$$\boxed{2\,1}\,, \quad \begin{smallmatrix}\boxed{3\,1}\\\boxed{1}\end{smallmatrix}, \quad \begin{smallmatrix}\boxed{3}\\\boxed{2}\\\boxed{1}\end{smallmatrix}, \quad \begin{smallmatrix}\boxed{3\,2}\\\boxed{2\,1}\end{smallmatrix}, \quad \begin{smallmatrix}\boxed{4\,3\,1}\\\boxed{2\,1}\end{smallmatrix}, \tag{69}$$

and calculate the corresponding normalisation factors,

$$|\square| = 2\,, \quad |\boxplus| = 3\,, \quad \left|\begin{smallmatrix}\square\\\square\end{smallmatrix}\right| = 6\,, \quad |\boxplus\!\boxplus| = 12\,, \quad |\boxplus\!\boxplus\!\boxplus| = 24\,. \tag{70}$$

Young operators $Y_\Theta \in \mathscr{A}(S_n)$ corresponding to standard Young tableaux are primitive idempotents. For $\Theta, \vartheta \in \mathscr{Y}_n$ they satisfy $Y_\Theta Y_\vartheta = 0$, i.e. they are transversal, if the corresponding Young diagrams have different shapes. Unfortunately, for different Young tableaux of the same shape it can happen that $Y_\Theta Y_\vartheta \neq 0$ when $n > 4$.

In birdtrack notation we can draw Young operators, using partial (anti-)symmetrisers as introduced in Eq. (60), e.g.

$$
Y_{\boxed{1\,2\,3}} = \;\vcenter{\hbox{\includegraphics{}}} \;, \qquad\qquad Y_{\begin{smallmatrix}\boxed{1}\\\boxed{2}\\\boxed{3}\end{smallmatrix}} = \;\vcenter{\hbox{\includegraphics{}}} \;,
$$

$$
Y_{\begin{smallmatrix}\boxed{1\,2}\\\boxed{3}\end{smallmatrix}} = \frac{4}{3}\;\vcenter{\hbox{\includegraphics{}}}\;, \qquad Y_{\begin{smallmatrix}\boxed{1\,3}\\\boxed{2}\end{smallmatrix}} = \frac{4}{3}\;\vcenter{\hbox{\includegraphics{}}}\;. \tag{71}
$$

Note that the normalisation factors are in agreement with Eq. (68) and the normalisation (57) of (anti-)symmetrisers. The following 5-box examples illustrate the loss of transversality for $n > 4$,

$$
Y_{\begin{smallmatrix}\boxed{1\,2\,3}\\\boxed{4\,5}\end{smallmatrix}} = 2\;\vcenter{\hbox{\includegraphics{}}}\;, \qquad \text{and} \qquad Y_{\begin{smallmatrix}\boxed{1\,3\,5}\\\boxed{2\,4}\end{smallmatrix}} = 2\;\vcenter{\hbox{\includegraphics{}}}\;, \tag{72}
$$

as we have

$$
Y_{\begin{smallmatrix}\boxed{1\,3\,5}\\\boxed{2\,4}\end{smallmatrix}} Y_{\begin{smallmatrix}\boxed{1\,2\,3}\\\boxed{4\,5}\end{smallmatrix}} = 0 \qquad \text{but} \qquad Y_{\begin{smallmatrix}\boxed{1\,2\,3}\\\boxed{4\,5}\end{smallmatrix}} Y_{\begin{smallmatrix}\boxed{1\,3\,5}\\\boxed{2\,4}\end{smallmatrix}} \neq 0\,. \tag{73}
$$

**Exercise 16** *Verify Eq. (73).*

### 3.3 Young operators and SU($N$): multiplets

Recall that $V \cong \mathbb{C}^N$ (or $\overline{V}$) carries the defining (or complex conjugate) representation of SU($N$). A tensor product $V^{\otimes n}$ (or $\overline{V}^{\otimes n}$) carries a product representation of SU($N$). This tensor product also carries a representation of $S_n$. The representations of these two groups commute, since SU($N$) acts only on individual factors, on each in the same way, whereas $S_n$ acts by permuting the factors. In fact, it is a standard result, that Young operators viewed as linear maps $V^{\otimes n} \to V^{\otimes n}$ (or $\overline{V}^{\otimes n} \to \overline{V}^{\otimes n}$) project onto irreducible SU($N$)-invariant subspaces.

In birdtrack notation this means that we simply add arrows to the lines in all diagrams for Young operators, all pointing in the same direction, e.g., $Y_{\scriptsize\young(12,3)} : V^{\otimes n} \to V^{\otimes n}$ reads

$$Y_{\scriptsize\young(12,3)} = \frac{4}{3}\ \raisebox{-1em}{[birdtrack diagram]}\ , \tag{74}$$

and since all arrows point in the same direction, we usually immediately drop them again.

We refer to irreducible SU($N$)-invariant subspaces as multiplets, e.g., a one-dimensional subspace (carrying the trivial representation) is called singlet. The dimension of a multiplet is given by the trace of the projector, e.g.

$$\operatorname{tr} Y_{\scriptsize\young(12)} = \raisebox{-1.5em}{[birdtrack diagram]}$$

$$= \frac{1}{2}\left( \raisebox{-1em}{[diagram]} + \raisebox{-1em}{[diagram]} \right)$$

$$= \frac{1}{2}(N^2 + N)$$

$$= \frac{N(N+1)}{2}, \tag{75}$$

or

$$\operatorname{tr} Y_{\scriptsize\young(12,3)} = \frac{4}{3}\ \raisebox{-1.5em}{[birdtrack diagram]}$$

$$= \frac{2}{3}\left( \raisebox{-1.5em}{[diagram]} - \raisebox{-1.5em}{[diagram]} \right)$$

$$= \frac{1}{3}\left( N^2(N+1) - N(N+1) \right)$$

$$= \frac{N}{3}(N^2 - 1), \tag{76}$$

For $N = 3$ the latter describes an octet, which actually carries the adjoint representation of SU(3).

**Exercise 17** *The Young diagram for the adjoint representation of* SU($N$) *is given by a column of* $N-1$ *boxes and a column with one box – why? Verify that the dimension of the adjoint representation is* $N^2-1$ *by calculating* tr $Y$ *in birdtracks.*

*Hint: First expand the symmetriser, then evaluate the trace of the remaining anti-symmetriser using the recursion relation* (66).

## 4 Colour space

Consider a QCD process with a couple of incoming and outgoing quarks, anti-quarks, and gluons. A corresponding Feynman diagram might look like this,

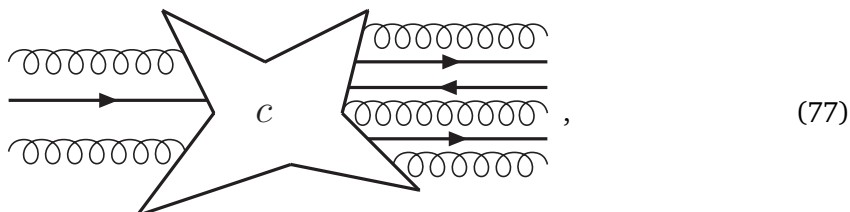

(77)

where inside some of the lines are connected directly, whereas others are tied together according to the QCD Feynman rules, i.e. by quark-gluon vertices as well as by 3-gluon and 4-gluon vertices. Note that these rules ensure that there are always as many (anti-)quark lines with arrows pointing away from the central blob as there are lines with arrows pointing towards it. The amplitude corresponding to such a diagram is the product of a kinematic factor (some often initially divergent momentum space integral), a spin factor and a colour factor. Here, we are only interested in the colour factor, the so-called colour structure.

The colour structure is a tensor $c \in (\overline{V} \otimes V)^{\otimes n_q} \otimes A^{\otimes n_g}$. Since colour is confined the only relevant colour structures are singlets, i.e. tensors $c$ which transform in the trivial representation of SU($N$). In other words, $c$ is a so-called invariant tensor. The singlet-subspace of $(\overline{V} \otimes V)^{\otimes n_q} \otimes A^{\otimes n_g}$ is called colour space.

Quark lines , gluon lines  and quark-gluon vertices (generators)  are examples of invariant tensors. Any combination (tensor product, contraction) of invariant tensors is also an invariant tensor. Thus, Young operators and triple gluon vertices (structure constants),

$$\text{(birdtrack)} = \frac{2}{T_R} \text{(birdtrack)} , \tag{78}$$

cf. Eq. (38), are invariant tensors, and also the fully symmetric

$$\text{(birdtrack)} := \frac{2}{T_R} \text{(birdtrack)} . \tag{79}$$

The vector space $V \cong \mathbb{C}^N$, which carries the defining representation of SU($N$), is endowed with a scalar product, which is also left invariant by SU($N$). This scalar product also induces scalar products on $\overline{V}$, $A$ and on arbitrary tensor products of $V$, $\overline{V}$ and $A$. For two colour structures $c_1, c_2 \in (\overline{V} \otimes V)^{\otimes n_q} \otimes A^{\otimes n_g}$ we have

$$\langle c_1, c_2 \rangle = \text{tr}(c_1^\dagger c_2) = \left( ((c_1)^{ab\dots}{}^{jk\dots}{}_{\ell m\dots})^* ((c_2)^{ab\dots}{}^{jk\dots}{}_{\ell m\dots} , \tag{80}$$

where in birdtrack notation Hermitian conjugation corresponds to mirroring the diagram across a vertical line and reversing all arrows.

For calculations it is convenient to expand colour structures into a basis of colour space. The most popular bases in use are so-called trace bases (which in general are overcomplete, i.e. they are no proper bases but only spanning sets). In general, trace bases are not orthogonal. In the remainder of this section, after a brief review of trace bases, we will discuss the construction of minimal, orthogonal bases, so-called multiplet bases.

## 4.1 Trace bases vs. multiplet bases

**Trace bases**

For diagrams with a given number of external (anti-)quark lines and gluon lines, the trace basis can be constructed as follows:

- Attach a quark-gluon vertex to each external gluon line.

- Then connect all external (anti-)quark lines to either external(anti-)quark lines or (anti-)quark lines from quark-gluon vertices.

Examples:

For $\overline{V} \otimes V \otimes A^{\otimes 2}$ we have to connect the (anti-)quark lines ending on the dashed box,

$$
\text{(81)}
$$

in all possible ways. The non-zero possibilities are

$$
\text{(82)}
$$

yielding

$$
c_1 = \qquad , \quad c_2 = \qquad , \quad c_3 = \qquad , \qquad \text{(83)}
$$

where we have omitted irrelevant prefactors. Note that in general the $c_j$ are not mutually orthogonal, e.g.

$$
\langle c_1, c_2 \rangle = \mathrm{tr}(c_1^\dagger c_2) = \qquad = T_R(N^2 - 1) = C_F N \, , \qquad \text{(84)}
$$

see Eqs. (42) and (43).

For $A^{\otimes 4}$, connecting the (anti-)quark lines ending on the dashed box,

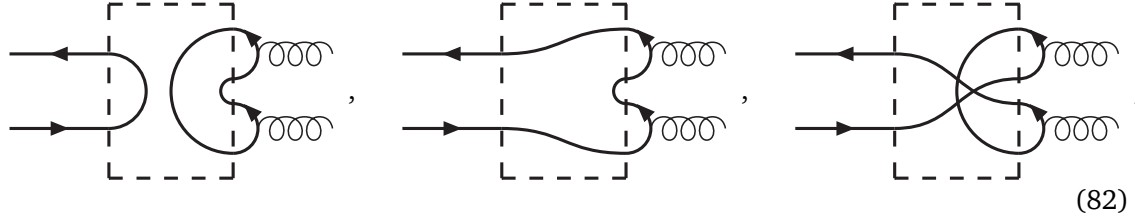

$$
\text{(85)}
$$

in all possible ways, we in turn find the non-vanishing diagrams

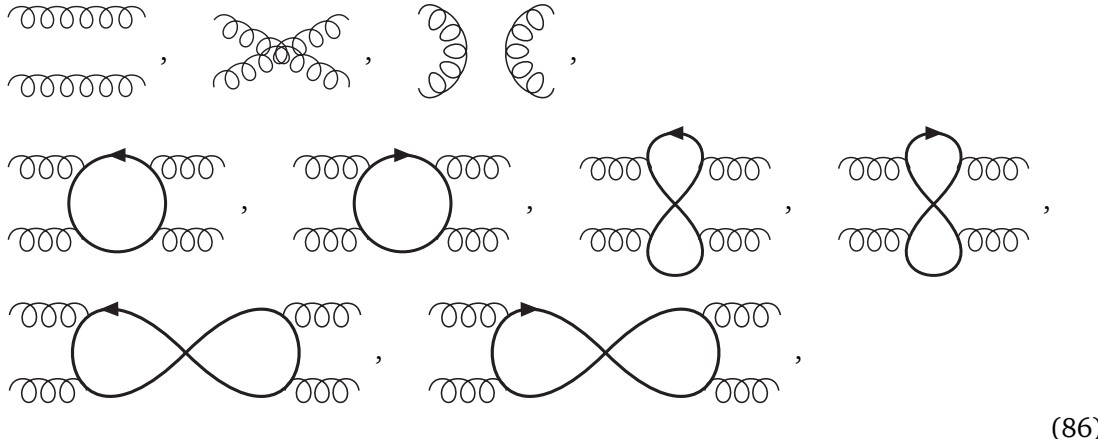

$$(86)$$

again omitting all prefactors. Written in index (or matrix) notation these basis vectors consist of traces of products of generators, thus the name trace basis. The colour structures (86) are again not mutually orthogonal, e.g.

$$\left\langle \begin{array}{c} \text{⬚} \end{array} , \begin{array}{c} \text{⬚} \end{array} \right\rangle = \begin{array}{c} \text{⬚} \end{array} = N^2 - 1 . \tag{87}$$

Moreover, for $N = 3$ the nine $A^{\otimes 4}$-vectors (86) are not linearly independent, since, as we will see below, the colour space which they span is only eight-dimensional.

There is a simple algorithm, for writing arbitrary colour factors as linear combinations of trace basis elements: First replace all four-gluon vertices by (one gluon-contracted linear combinations of) three-gluon vertices. Then replace all three-gluon vertices by (linear combinations of) quark loops with three gluons attached, see Eqs. (38) or (78). Finally, remove all internal gluon lines using Eq. (42).

**Multiplet bases**

For the construction of a multiplet basis for $c \in (\overline{V} \otimes V)^{\otimes n_q} \otimes A^{\otimes n_g}$ we consider $c$ as a linear map, say

$$c : (\overline{V} \otimes V)^{\otimes k_q} \otimes A^{\otimes k_g} \rightarrow (\overline{V} \otimes V)^{\otimes (n_q - k_q)} \otimes A^{\otimes (n_g - k_g)} , \tag{88}$$

for some $0 \leq k_q \leq n_q$ and $0 \leq k_g \leq n_g$. In general, we thus have a linear map $c : W_1 \rightarrow W_2$, between two vector spaces, carrying representations $\Gamma_1$ and $\Gamma_2$ of SU($N$). Moreover, $c$ being an invariant tensor means that

$$c \circ \Gamma_1(g) = \Gamma_2(g) \circ c \quad \forall g \in \text{SU}(N) . \tag{89}$$

Now we are in a situation where we can employ Schur's lemma. It is often formulated for the case where $W_1$ and $W_2$ carry irreducible representations saying that

- if the two representations are inequivalent, then $c$ vanishes identically, and
- if the two representations are equivalent and $W_1 = W_2$, then $c$ is a multiple of the identity.

In our case the representations are typically not irreducible, and then Schur's lemma implies that $c$ can only map subspaces onto each other that carry the same irreducible representation, i.e. $c$ maps only equivalent multiplets onto each other.

Now consider the case when $W_1 = W_2 =: W$. If we decompose $W$ into multiplets, i.e. into irreducible SU($N$)-invariant subspaces, then the projectors onto these multiplets are distinguished elements of colour space.

|  | Number of multiplets | | Dimension of colour space | |
|---|---|---|---|---|
|  | $N = 3$ | $N = \infty$ | $N = 3$ | $N = \infty$ |
| $A^{\otimes 2} \to A^{\otimes 2}$ | 6 | 7 | 8 | 9 |
| $A^{\otimes 3} \to A^{\otimes 3}$ | 29 | 51 | 145 | 265 |
| $A^{\otimes 4} \to A^{\otimes 4}$ | 166 | 513 | 3 598 | 14 833 |
| $A^{\otimes 5} \to A^{\otimes 5}$ | 1 002 | 6 345 | 107 160 | 1 334 961 |

Table 1: Number of projection operators and dimension of the colour space within $A^{\otimes(2n)}$, for colour structures viewed as maps $A^{\otimes n} \to A^{\otimes n}$. The first two columns show the number of multiplets (counted with multiplicities) in the decomposition of $A^{\otimes n}$, both for $N = 3$ and for $N \geq n$. The last two columns contain the dimensions of the respective colour spaces; the dimension in the last column is also equal to the number of elements of the corresponding trace basis.

- If each multiplet appears only once in the decomposition of $W$ then the projectors form a basis of colour base. If, moreover, the projectors are Hermitian, then this basis is orthogonal.

- If some multiplets in the decomposition of $W$ have a multiplicity $> 1$ then we have to complement the projectors with operators mapping equivalent multiplets onto each other.

In practice, finding the multiplets in the decomposition of $W$ and their multiplicities can, e.g., be done by multiplying Young diagrams according to the standard rules. The crucial step is then to find Hermitian projectors onto these multiplets. Finally, multiplet bases can be constructed straightforwardly from Hermitian projection operators.

In the following sections we discuss how to construct Hermitian projection operators as well as multiplet bases for the cases $V^{\otimes n} \to V^{\otimes n}$ and $A^{\otimes n} \to A^{\otimes n}$. Moreover, we will see that multiplet cases for any colour space can be constructed from projectors for $A^{\otimes n} \to A^{\otimes n}$.

**Comparison**

Trace bases are convenient since they are easy to construct and since there is a simple algorithm for expanding arbitrary colour factors into a trace basis. In general, trace bases are overcomplete, i.e. expansions tend to have too many terms. For instance, the trace basis for the colour space within $A^{\otimes n}$ is a proper basis if $n \leq N$ but for $n > N$ it is only a spanning set – the basis vectors are linearly dependent. Trace bases, typically, are also non-orthogonal.

Constructing multiplet bases requires more work than constructing trace bases. In return we obtain not only a proper basis, i.e. the basis vectors are linearly independent, but also an orthogonal basis. Even though the dimension of colour space depends on $N$, the number of colours, the birdtrack construction of multiplet bases can be carried out independently of $N$, and then for small $N$ some basis vectors simply vanish.

The numbers in Table 1 give us an impression of the potential advantage of multiplet bases over trace bases. Imagine doing a calculation for $N = 3$ with 6 to 10 external gluons involved. Then the number of trace basis elements exceeds the dimension of colour space by, roughly, a factor of 2 to 12. Expanding colour structures in a trace or multiplet basis and then, e.g., calculating scalar products will result in 4 to 144 times as many terms when using a trace basis instead of a multiplet basis.

### 4.2 Multiplet bases for quarks

We first consider the case without external gluons, i.e. we are interested in the colour space within $(\overline{V} \otimes V)^n$. Tensors $c \in (\overline{V} \otimes V)^{\otimes n}$ can be viewed as linear maps $c : V^{\otimes n} \to V^{\otimes n}$, and

Young operators $Y_\Theta$ project onto multiplets. Unfortunately, Young operators are in general not Hermitian, as can be seen by, e.g. inspecting Eq. (74): Mirroring and reversing the arrows does not yield back the original expression.

However, Hermitian operators $P_\Theta$ corresponding to standard Young tableaux $\Theta$ can be constructed. In [1] they are derived as solutions of certain characteristic equations. They can also be written down directly starting from a Young tableaux as follows. Consider the sequence of Young tableaux $\Theta_j \in \mathscr{Y}_j$ obtained from $\Theta \in \mathscr{Y}_n$ by, step by step, removing the box with the highest number, e.g., starting with $\Theta = \Theta_3 = \boxed{\begin{smallmatrix}1&2\\3&\end{smallmatrix}} \in \mathscr{Y}_3$ we obtain

$$\Theta_1 = \boxed{1}, \quad \Theta_2 = \boxed{1\,2}, \quad \Theta_3 = \boxed{\begin{smallmatrix}1&2\\3&\end{smallmatrix}}. \tag{90}$$

Young operators for $n = 2$ are Hermitian – they are just total (anti-)symmetrisers – so we set

$$P_{\Theta_j} = Y_{\Theta_j} \quad \forall\, j \le 2. \tag{91}$$

Then we define recursively

$$P_{\Theta_j} = (P_{\Theta_{j-1}} \otimes \mathbb{1}_V) Y_{\Theta_j} (P_{\Theta_{j-1}} \otimes \mathbb{1}_V) \quad \forall\, j \ge 3, \tag{92}$$

i.e. in birdtrack notation we take the Young operator $Y_{\Theta_j}$ and write the Hermitian Young operator $P_{\Theta_{j-1}}$ over the first $j-1$ lines, to the left and to the right. For instance,

$$P_{\boxed{\begin{smallmatrix}1&2\\3&\end{smallmatrix}}} = \frac{4}{3}\; \text{[birdtrack]} = \frac{4}{3}\; \text{[birdtrack]}, \tag{93}$$

which is manifestly Hermitian, and in birdtracks it is also easy to see that

$$\operatorname{tr} P_{\boxed{\begin{smallmatrix}1&2\\3&\end{smallmatrix}}} = \operatorname{tr} Y_{\boxed{\begin{smallmatrix}1&2\\3&\end{smallmatrix}}} \tag{94}$$

since $\left(\text{[birdtrack]}\right)^2 = \text{[birdtrack]}$.

It can be shown [3] that the resulting $P_\Theta$ not only project onto the correct multiplets but that they are also Hermitian and thus mutually orthogonal with respect to the scalar product (80). Furthermore, using the Hermitian Young operators $P_\Theta$ automatically cures the loss of transversality mentioned at the end of Sec. 3.2

The recursive construction can produce initially lengthy expressions which can often be simplified considerably, see, e.g., the step-by-step example for $\boxed{\begin{smallmatrix}1&3&5\\2&4&\end{smallmatrix}}$ in the Appendix of [3]. Similar simplifications can be shown to occur much more generally [4] and they can be used to devise a recipe for directly writing down fully simplified Hermitian Young operators [5].

With the Hermitian Young operators

$$P_{\boxed{1\,2\,3}} = \text{[birdtrack]}, \quad P_{\boxed{\begin{smallmatrix}1&2\\3&\end{smallmatrix}}} = \frac{4}{3}\,\text{[birdtrack]}, \quad P_{\boxed{\begin{smallmatrix}1&3\\2&\end{smallmatrix}}} = \frac{4}{3}\,\text{[birdtrack]}, \quad P_{\boxed{\begin{smallmatrix}1\\2\\3\end{smallmatrix}}} = \text{[birdtrack]}, \tag{95}$$

we have completely decomposed $V^{\otimes 3}$ into an orthogonal sum of multiplets. However, the $P_\Theta$ alone do not form a basis for the colour space within $(\overline{V} \otimes V)^{\otimes 3}$, since the multiplet $\boxminus$ appears twice, i.e. we also need an operator mapping these two multiplets onto each other. To this end we write down $P_{\boxed{\begin{smallmatrix}1&2\\3&\end{smallmatrix}}}$ and $P_{\boxed{\begin{smallmatrix}1&3\\2&\end{smallmatrix}}}$ next to each other (omitting prefactors),

$$\text{[birdtrack]}, \tag{96}$$

and seek a way of connecting the lines within the dashed box such that the whole expression does not vanish, because then it is guaranteed, that the resulting expression has the same kernel as $P_{\tiny\begin{array}{cc}1&3\\2&\end{array}}$ and the same image as $P_{\tiny\begin{array}{cc}1&2\\3&\end{array}}$. The only such connection (up to a sign) is

$$\text{(97)}$$

and by expanding the central (anti-)symmetrisers one can verify that this expression is proportional to

$$T_1 := \qquad\qquad . \qquad\qquad\qquad\qquad (98)$$

Thus, we have found a basis vector mapping $\tiny\begin{array}{cc}1&3\\2&\end{array}$ to $\tiny\begin{array}{cc}1&2\\3&\end{array}$. The vector for the reverse mapping can be obtained in the same way and reads

$$T_2 := \qquad\qquad . \qquad\qquad\qquad\qquad (99)$$

**Exercise 18** *Define*

$$B = \qquad\qquad , \qquad\qquad\qquad\qquad (100)$$

*and show that $B^2$ is proportional to $B$ by expanding the central (anti-)symmetrisers. Explain why this implies that the birdtrack diagram (97) is proportional to $T_1$.*

The multiplet basis for $V^{\otimes 3} \to V^{\otimes 3}$ consisting of four Hermitian Young operators and two transition operators is orthogonal. If desired the basis vectors can be normalised: For Hermitian projection operators we generally have

$$\langle P_\Theta, P_\Theta \rangle = \text{tr}(P_\Theta^\dagger P_\Theta) = \text{tr}(P_\Theta P_\Theta) = \text{tr}(P_\Theta) = \dim M_\Theta \qquad (101)$$

where $M_\Theta$ is the multiplet to which $P_\Theta$ projects. Hence,

$$\frac{1}{\sqrt{\dim M_\Theta}} P_\Theta \qquad\qquad\qquad\qquad (102)$$

has norm one. The transition operators can straightforwardly be normalised by a direct calculation.

**Exercise 19** *Calculate $\langle T_1, T_1 \rangle$ and normalise $T_1$ accordingly.*

## 4.3 General multiplet bases

**Gluons only**

The construction of multiplet bases for an even number of external gluons has been outlined in Sec. 4.1: Consider colour structures as maps $A^{\otimes n} \to A^{\otimes n}$, construct Hermitian projection

operators to multiplets within $A^{\otimes n}$, and complement the projection operators with transition operators between equivalent multiplets.

For an odd number of external gluons, i.e. when considering the colour space within $A^{\otimes(2n+1)}$ we consider colour structures as maps $A^{\otimes n} \to A^{\otimes(n+1)}$. In Sec. 4.4 we will see that knowing the projectors to multiplets within $A^{\otimes n}$ allows to construct the projectors to all equivalent multiplets within $A^{\otimes(n+1)}$ in a straightforward way.[4] Transition operators between equivalent multiplets within $A^{\otimes n}$ and $A^{\otimes(n+1)}$ then constitute the desired multiplet basis.

### Quarks and gluons

When (anti-)quark lines are present we can always group together an anti-quark line with a quark line. Consider, e.g., the colour space within $\overline{V} \otimes V \otimes A^{\otimes n}$. Noting that $\overline{V} \otimes V = \bullet \oplus A$, see Eq. (41), the quark-anti-quark pair can be either in a singlet state or in a state transforming in the adjoint representation. In the singlet case we have to study the colour space within $A^{\otimes n}$, in the adjoint case the colour space within $A^{\otimes(n+1)}$.

### Conclusion

General multiplet bases can be constructed in a straightforward way from gluon projectors. In particular, the gluon projectors for $A^{\otimes \nu} \to A^{\otimes \nu}$, $\nu = 0, \ldots, n$, are sufficient for constructing multiplet bases for the colour spaces within $\left(\overline{V} \otimes V\right)^{\otimes k} \otimes A^{\otimes(2n+1-k)}$ with arbitrary $k = 0, \ldots, 2n+1$.

## 4.4 Gluon projectors

In Sec. 4.3 we have seen that the crucial ingredient for any multiplet basis are the projection operators to multiplets within $A^{\otimes n}$.

The construction rules for projectors depend on when a multiplet $M$ appears for the first time in the sequence

$$A^{\otimes 0} = \bullet, \ A^{\otimes 1} = A, \ A^{\otimes 2} = A \otimes A, \ A^{\otimes 3}, \ A^{\otimes 4}, \ldots \tag{103}$$

We call $n_f(M) = 0, 1, 2, 3, 4, \ldots$ the first occurrence of multiplet $M$. Consequently, the only multiplets with first occurrence 0 and 1 are the trivial and the adjoint representation, respectively, in short $n_f(\bullet) = 0$ and $n_f(A) = 1$. For SU(3), we have, e.g.,

$$
\begin{array}{cccccccccccccc}
\square & \otimes & \square & = & \bullet & \oplus & \square & \oplus & \square & \oplus & \square\square\square & \oplus & \boxed{} & \oplus & \square \\
8 & & 8 & & 1 & & 8 & & 8 & & 10 & & \overline{10} & & 27
\end{array}
\tag{104}
$$

and, consequently, the decuplets and the 27-plet have first occurrence 2. The following table shows some more SU(3) examples.

| $n_f$ | 0 | 1 | 2 | 3 |
|---|---|---|---|---|
| SU(3) Young diagrams | $\bullet = \square$ | $A = \square$ | | |

[4] Using the terminology of Sec. 4.4 only projectors to old multiplets have to be constructed.

The first occurrence of any multiplet can in principle be determined by repeatedly multiplying Young diagrams for the adjoint representation until the desired multiplet appears. One can also derive [6, App. B] a graphical rule for directly determining $n_f(M)$ from the corresponding Young diagram.

Our construction of gluon projectors will be recursive. Assume that we have determined the projectors for the decomposition of $A^{\otimes(n-1)} = \bigoplus_j M_j$ into multiplets $M_j$. In order to decompose $A^{\otimes n}$ we have to multiply each $M \subseteq A^{\otimes(n-1)}$ with another $A$, i.e. we consider

$$M \otimes A = \bigoplus_k M'_k, \tag{105}$$

which for the projectors reads

$$\text{(diagram)} \tag{106}$$

As for the Hermitian Young opeators, cf. the discussion around Eqs. (90)–(92), our projectors will be such that $\forall\, k$

$$\text{(diagram)} \qquad \text{is proportional to} \qquad \text{(diagram)}. \tag{107}$$

For the decomposition (105) one can show [6] that

(i) $n_f(M'_k) = n_f(M)-1$, $n_f(M)$ or $n_f(M)+1$, and

(ii) only $M$ itself can appear with multiplicity greater than one within $M \otimes A$ (in fact it can appear up to $N-1$ times), all other multiplets are unique.

Reexamining Eq. (104), where on the l.h.s. we identify $\boxbar \otimes \boxbar = M \otimes A$, we can verify both statements: Property (i) is trivially true since $n_f(M) = 2$, but we also see that (ii) holds, as on the r.h.s. all multiplets except for $M = \boxbar$ appear only once, and $M = \boxbar$ itself appears twice (which here is the maximum degeneracy since $N = 3$).

We call a multiplet $M \subseteq A^{\otimes n}$ old if $n_f(M) < n$ and new if $n_f(M) = n$. Construction rules for projectors onto multiplets $M'_k \subseteq M \otimes A$ depend on whether $M$ and $M'$ are old or new and on which of the cases (i) we have at hand. We now give some examples for important cases, which all appear in $A \otimes A$; the complete set of construction rules (and their proofs) are given in [6].

### $M$ new, $n_f(M'_k) = n_f(M)-1$

Write down $P_M$ twice and bend back the last gluon line,

$$P_{M'_k} = \frac{\dim M'_k}{\dim M} \;\; \text{(diagram)} \;\; ; \tag{108}$$

the prefactor makes sure that $P_{M'_k}^2 = P_{M'_k}$. Example: The projector to the singlet in eq. (104) is constructed in this way,

$$P_\bullet = \frac{1}{N^2 - 1} \quad .\tag{109}$$

### $n_f(M'_k) = n_f(M)$, $M'_k$ equivalent to $M$

Write down $P_M$ twice and and connect the new gluon line to one of the old gluon lines by means of two $f$- or $d$-vertices (or a linear combination of them),

$$P_{M'_k} = \gamma \quad \boxed{P_M} \quad \boxed{P_M} \quad \boxed{P_M} \quad ,\tag{110}$$

where $\otimes$ is to be replaced by $\bullet$ or $\circ$ (or a linear combination). A formula for the normalisation factor $\gamma$ is given in [6]. Examples: The two projectors to copies of the adjoint representation on the r.h.s. of Eq. (104) are constructed in this way,

$$P_{Aa} = \frac{1}{2N T_R} \quad \text{and} \quad P_{As} = \frac{N}{2(N^2 - 4)T_R} \quad .\tag{111}$$

### New multiplets

Here we use that $A \subset \overline{V} \otimes V$: Split each gluon line into a quark and an anti-quark line by means of a generator (quark-gluon-vertex). Then put a Young operator (Hermitian or not doesn't matter) on the quark lines and on the anti-quark-lines (one each),

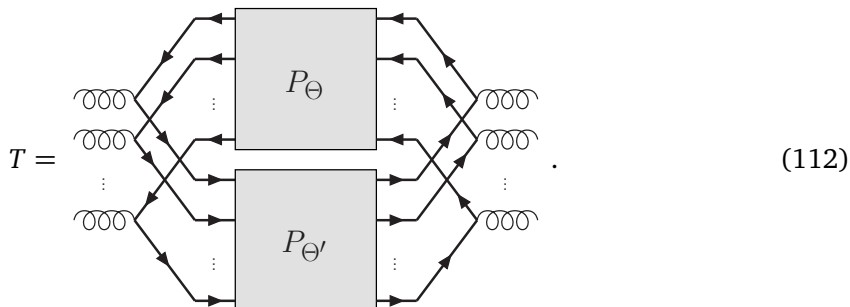

$$T = \tag{112}$$

One can show [6] that for $\Theta, \Theta' \in \mathscr{Y}_n$ the tensor product $\overline{\Theta} \otimes \Theta'$ contains exactly one new multiplet of the decomposition of $A^{\otimes n}$. Since it also contains contributions from other (old) multiplets, these have to be removed in a Gram-Schmidt step,

$$\widetilde{T} = T - \sum_{M \text{ old}} \frac{\text{tr}(P_M T)}{\dim M} P_M .\tag{113}$$

Finally, we obtain the desired projector onto the new multiplet by normalising $\widetilde{T}$,

$$P_{M'_k} = \frac{\dim M'_k}{\text{tr} \widetilde{T}} \widetilde{T} .\tag{114}$$

**Examples**

Choosing ($\square\square$, $\square\square$) for the pair $(\Theta, \Theta')$ this procedure leads to the projector onto the 27-plet in Eq. (104). Projectors onto the two decuplets in Eq. (104) we find by choosing ($\square\square$, $\boxminus$) and ($\boxminus$, $\square\square$). The resulting formulae are, e.g., given in [1, Table 9.4], [7, App. A.1] or [6, Eq. (1.23)].

Note that all birdtrack construction rules in this section never use that $N = 3$. Thus, the projection operators constructed above, project onto multiplets within $A \otimes A$ for any $N$. However, for $N \geq 4$ there is exactly one more multiplet in the decomposition of $A \otimes A$. We find the corresponding projector by applying the construction rules for new multiplets, this time using $(\Theta, \Theta') = (\boxminus, \boxminus)$; the result can be shown to vanish for $N = 3$. This is a general feature of these birdtrack constructions for gluon projectors (and for multiplet bases): The construction rules are independent of $N$. If for small $N$ there are fewer multiplets (or colour spaces of smaller dimension) then some of the terms simply vanish – as opposed to less obvious linear dependencies, which appear in trace bases, cf. Eq. (86).

**Exercise 20** *Construct the projectors $P_{\square\square\square}$, $P_{\boxplus}$ and $P_{\boxplus\square}$ according to the rules given above. You may either use $\overline{P_{\square\square\square}} = P_{\boxplus}$ in your calculations or verify this property from your result. Also determine the dimensions of the corresponding multiplets for arbitrary $N$.*

## 4.5 Some multiplet bases

**$A^{\otimes 4}$**

The multiplet basis for the colour space within $A^{\otimes 4}$ is given by the projection operators – six for $N = 3$, seven for $N \geq 4$ – normalised according to Eq. (102), and two transition operators mapping the two multiplets carrying the adjoint representation onto each other. The latter we construct by writing down projectors for each of the two copies (ignoring prefactors),

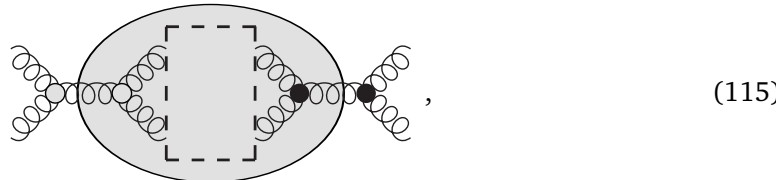

$$\text{(115)}$$

and then seeking a non-vanishing connection inside the dashed rectangular box. We do not have to explicitly write out such a connection; simply notice that when we have found one then the expression inside the grey ellipse is an invariant tensor mapping $A$ to $A$ and thus, according to Schur's lemma, it is proportional to $\text{〜〜〜}$. Hence, the first transition operator is proportional to

$$\text{(116)}$$

We obtain the second transition operator by interchanging $\bullet$ and $\circ$.

**Exercise 21** *Normalise the transition operator* (116).

$\overline{V} \otimes V \otimes A^{\otimes 2}$

The quark-anti-quark pair can either be in a singlet state or in the adjoint representation. If it is in a singlet state we need to find a transition operator mapping the singlet within $A \otimes A$ to the singlet within $\overline{V} \otimes V$. To this end we write down the two projectors (ignoring prefactors) and seek a non-vanishing connection inside the dashed box,

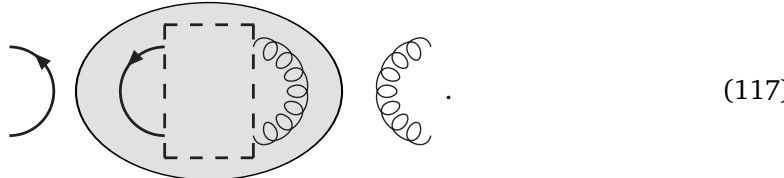

(117)

No matter what this connection looks like, the part inside the grey ellipse is just a number, i.e. the desired transition operator is proportional to

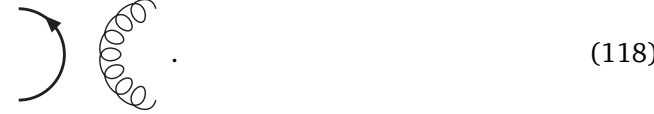

(118)

**Exercise 22** *Find a non-vanishing way to connect the lines withing the dashed box in diagram (117) and evaluate the resulting term inside the grey ellipse.*

If the quark-anti-quark pair is in the adjoint representation, then we need to find transition operators to the two adjoint representations within $A \otimes A$. Once more we write down the corresponding projectors (omitting prefactors), and seek non-vanishing connections inside the dashed boxes,

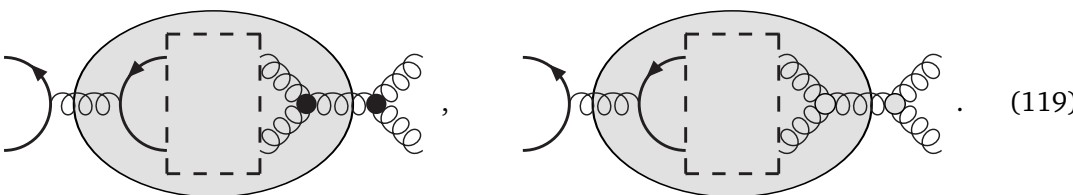

(119)

Once more, we do not have to find these connections explicitly, but simply notice that the parts within the grey ellipses have to be proportional to 〜, i.e. the transition operators are proportional to

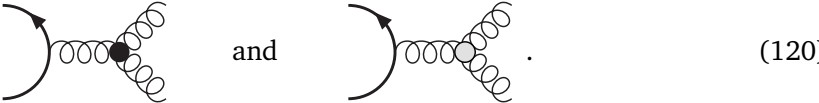

and

.

(120)

The three tensors in (118) and (120) form an orthogonal multiplet basis for the colour space within $\overline{V} \otimes V \otimes A^{\otimes 2}$, which is to be compared to the non-orthogonal trace basis (83).

**Exercise 23** *Normalise the basis vectors in (118) and (120).*

# 5 Further reading

I give some recommendations for where to learn more about the topics touched upon in these lectures. The list is by no means complete. If you are interested in the history of birdtracks, I recommend Sec. 4.9 of THE BOOK.

**General introductions to and compendiums for birdtracks**

- THE BOOK on birdtracks is Predrag Cvitanović's *Group Theory: Birdtracks, Lie's and Exceptional Groups* [1]. You want to have it next to you whenever you do birdtrack calculations. A precursor are Cvitanović's 1984 lecture notes [8] which already contain a lot of the material covered in THE BOOK. The 1976 paper [9] provides a nice introduction and summary of birdtracks for SU($N$) and QCD.

- Roger Penrose developed birdtracks to be used for tensors in general relativity. You get a nice impression from his semi pop-science book *The Road to Reality* [2]. There, birdtrack diagrams are introduced in Secs. §12.8, §13.3–§13.9, §14.3, §14.4, §14.6, and §14.7 in the context of differential geometry and later (Secs. §19.2, §19.6, §22.12, §26.2, §29.5) used for general relativity and other topics.

- The book *Diagram Techniques in Group Theory* by Geoffrey E. Stedman [10] also treats many aspects and applications of the the birdtrack method; it also contains an introductory section on vector algebra.

- Yuri L. Dokshitzer's lecture notes *Perturbative QCD (and beyond)* [11] introduce birdtrack techniques using QCD processes as examples; some of his notational conventions differ slightly differ from ours.

- If you are looking for a cheat sheet on birdtracks I suggest App. A of my paper [6] with Malin Sjödahl.

**Background**

Classic results on the representation theory of finite groups (such as $S_n$) and compact Lie groups (such as SU($N$)), on Schur's lemma, or on the multiplication of Young diagrams can, e.g., be found in [12–14] and in a vast number of other textbooks.

**Details on the specific topics (of later sections) of this course**

- Young operators (non-Hermitian) in birdtracks are discussed in [15].

- The main references for the construction of multiplet bases are [3,6]. Hermitian Young operators are constructed in [3], gluon projectors and the general rules for constructing multiplet bases are derived in [6].

  Hermitian Young operators and multiplet basis for few quarks appear already in [1,16, 17]. The multiplet basis for $A^{\otimes 2} \to A^{\otimes 2}$ in birdtracks is constructed in [1] using a different method.

  Simplification rules for a more efficient construction of Hermitian Young operators are derived in [4,5]. The corresponding quark multiplet bases are discussed in [18]

- When pen and paper calculations become unwieldy, the Mathematica and C++ packages [19,20] by Malin Sjödahl come in handy.

- Sjödahl and co-workers discuss decomposition into multiplet bases [21] and recursion relations [22].

**Typesetting**

All birdtracks in these notes where drawn with `JaxoDraw` [23].

**Epilogue**

On the virtue of different notations consider the following three cartoons[5] from abstrusegoose.com:

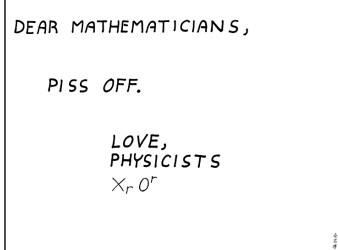

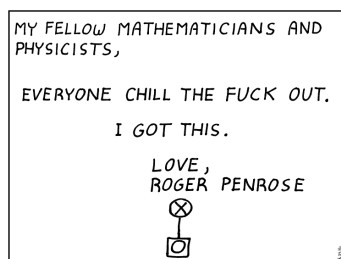

http://abstrusegoose.com/128          http://abstrusegoose.com/129          http://abstrusegoose.com/130

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
