# Peer review of "Birdtracks for SU(N)"

_SciPost Physics Lecture Notes, doi:SciPost Phys. Lect. Notes 3 (2018)_

## Round 1 · Referee Report · Anonymous · 2017-8-30

Strengths

1) The paper is a pedagogical, self-contained review article, accessible to a non-expert audience.
2) The text is well-written and clear, with very few typos.
3) It contains easy exercises that keep the reader active and interested.

Weaknesses

1) The paper lacks a proper introduction.

Report

Birdtrack notations find many applications in mathematics and physics, from the classification of Lie algebras to the computation of amplitudes in QCD. This paper gives a review of this subject for SU(N). The concepts are introduced pedagogically using simple examples and exercises, making it easy and enjoyable to read. I have a very favorable opinion of this manuscript and will recommend that it be accepted after the author adds an introduction.

Requested changes

1) The paper would highly benefit from the addition of a (short) introduction, reviewing of the historical development of the birdtrack notations and including comments on their present and future applications.

2) Is the tensor product form for $P_A$ in (39) correct? This would suggest that $(P_A)^2 = \frac1{(T_R)^2} (t^a t^b \otimes t^a t^b)$. However, if I am not mistaken, to compute $(P_A)^2 = P_A$ using the index notation, one should proceed as follows:

\begin{equation}((P_A)^2)^{j \ell}_{ k m} = \frac1{(T_R)^2} (t^a)^j_k (t^a)^p_q (t^b)^q_p (t^b)^\ell_m = \frac1{T_R} (t^a)^j_k (t^b)^\ell_m \delta^{ab} = \frac1{T_R} (t^a)^j_k (t^a)^\ell_m = (P_A)^{j \ell}_{ k m}\end{equation}

The factors $t^a$ and $t^b$ therefore do not combine as $t^a t^b \otimes t^a t^b$, as suggested by the tensor product form.

3) In eq. (108) and (110), the dependence upon $M’_k$ appears only in the choice of the overall constant. Is this correct? In (108) for instance, I instead expected that the birdtrack diagram would include projectors $P_{M’_k}$ instead of $P_M$.

  • validity: high
  • significance: good
  • originality: ok
  • clarity: top
  • formatting: perfect
  • grammar: perfect

Author:  Stefan Keppeler  on 2017-09-08  [id 167]

(in reply to Report 1 on 2017-08-30)
Category:
answer to question

Thank you very much for the careful report.

@2: You are right.
We have $t^a: V \to V$, and thus $\frac{1}{T_R} t^a \otimes t^a: V \otimes V \to V \otimes V$, whereas $P_A : \overline{V} \otimes V \to \overline{V} \otimes V$.
$P_A$ is defined by the components of $\frac{1}{T_R} t^a \otimes t^a$, but the equation to which you refer - the left part of Eq. (39) - is wrong. It's probably best to omit it and instead define $P_A$ directly in terms of its components (right part of Eq. (39)).

@3:
Eq. (108) is correct - essentially since we construct the projector onto $M_k' \subset M_k' \otimes A \otimes A$ and due to the general rule described in Eq. (107).
In Eq. (110) there is indeed a $P_{M_k'}$ missing in the middle (between the two $\otimes$-vertices).

By the way, there should have been a couple of $\;\vdots\;$ in the diagrams of Eqs. (106), (107), (108), and (110).

---

## Round 1 · Referee Report · Anonymous · 2017-9-7

Strengths

1. The paper gives a nice introduction to the use of Birdtracks for QCD applications.
2. It explains the construction of Hermitian projection operators onto the irreducible representations of SU(N).

Weaknesses

1. The paper switches midway from a general introduction to birdtrack tecniques to detailed application to trace and multiplet bases.

Report

Apart from many trivial corrections (see below), my main observations about this paper are that I am left unclear as to whether this is an introduction to birdtrack methods, or a recipe for how to use them to construct a basis to QCD diagrams. In particular, it does not explain why we should not just project all external legs into irreps, the coefficients being $3n-j$ Wigner coefficients which may be expressed in terms of $0$, $3$, and $6-j$ coefficients? (As in "THE BOOK"). What I am looking for here is some discussion of the computational complexity of the method presented, and why it is preferable to the "more general" $3n-j$ simplification algorithm.

I am also left wondering about the connection between the Hermitian projection operators for the subspaces carrying irreps. of SU(N) and those for the symmetric group. We were led to considering Young projectors corresponding to Young diagrams on the basis that these project irreps. of the symmetric group, and from these projectors one can construct the explicit representation matrices for the symmetric group. These matrices are not unitary, and the construction of unitary representations requires some more work (in principle this is just an application of Maschke's theorem, but in practice that is very lengthy). Moreover the unitary irreps. of the symmetric group are not matrices with integer (rational) entries, but require the introduction of some algebraic extensions. We might not care about the irreps. of the symmetric group here, of course, but it would be nice to understand the relation between them and the irreps. of SU(N).

Requested changes

1. On page 4 (and elsewhere) the notation $i\neq j\neq m\neq i$ is in principle ambiguous, since inequality (unlike equality) is not associative. I would suggest writing something like "$i$, $j$, and $m$ must be (pairwise) disjoint". Of course, it is obvious what is meant, so this is just nitpicking.
2. Below (16) either write $\sum_{j=1}^3 \delta_{jj} = 3$ or say explicitly at least once that the summation convention is being used.
3. On page 7 change "be it's dual" to "be its dual".
4. Above (30) change "Elements of the Lie algebra are called generators" to "A set of generators of the Lie algebra that form a basis as called generators" or similar.
5. Top of page 17. Why is it "unfortunate" that different Young tableaux of the same shape can have $Y_\Theta Y_\theta \neq 0$ when $n > 4$?
6. In (76) the final diagram is correct, but it is not the obvious way of drawing the diagram.
7. Below (77) "Feynamn" to "Feynman".
8."(some initially divergent momentum space integral)": surely they are not necessarily divergent.
9. "In other words, $c$ is a so-called invariant tensor": this usually just means a tensor that is invariant under the action of the Lie group SU(N), or equivalently the "Ward identity" for the Lie algebra su(N). It need not be a scalar.
10. "Color" to "colour", more nitpicking, but at least be consistent whether the paper is in English or American.
11. "tesnors" to "tensors".
12. Top of page 20 "and inverting all arrows" to "and reversing all arrows". I don't know how to invert an arrow.
13. "i.e. they are no proper bases..." to "i.e., they are not proper bases..."
14. "minimal, orthogal bases" to "minimal, orthogonal bases".
15. Table 1. "The last row is also equal to..." should be "The last column is also equal to..." I believe.

  • validity: high
  • significance: good
  • originality: good
  • clarity: good
  • formatting: good
  • grammar: good

Author:  Stefan Keppeler  on 2018-04-16  [id 245]

(in reply to Report 2 on 2017-09-07)

Thank you very much for the detailed report. In particular, thanks for also pointing out typos, which I happily corrected.

I am left unclear as to whether this is an introduction to birdtrack methods, or a recipe for how to use them to construct a basis to QCD diagrams.

It's both, as stated in the abstract. I hope the new introduction helps to make this clear.

In particular, it does not explain why we should not just project all external legs into irreps, the coefficients being 3n−j Wigner coefficients...

Assuming that you refer to Chapter 5 of THE BOOK, I don't view the topics discussed in these lectures (in particular multiplet bases for colour space) as an alternative to dealing with recouplings in terms of Wigner 3j and 6j coefficients. In fact, multiplet bases and 3n-j coefficients can be used alongside each other when analysing colour structures, see e.g. Ref. 21 (arXiv:1507.03814).

I am also left wondering about the connection between the Hermitian projection operators for the subspaces carrying irreps of SU(N) and those for the symmetric group...

One could, of course, say a lot more about irreps of the symmetric group. But that's beyond the scope of these notes.

On the requested changes:

  1. The formula $i \neq j \neq m \neq i$ is not ambiguous. I don't know what it means to say that equality is associative - do you mean transitive? I know disjoint sets, but I don't know what it means for numbers to be disjoint. I prefer to not change this formula.

  2. Before Eq. (2) I write that the summation convention is used. I think this is the right place for this statement, since this is where I use the summation convention for the first time.

  3. Done.

  4. Changed to: "Elements of a basis of the Lie algebra are called generators"

  5. Knowing that these products vanish is convenient in many calculations. Not having this property is less convenient or "unfortunate".

  6. I changed how I draw that diagram.

  7. Done.

  8. I inserted "often".

  9. Being "invariant under the action of the Lie group SU(N)" is the same as "transform[ing] in the trivial representation of SU(N)". I am not sure whether there is a question or requested change.

  10. Done.

  11. Done.

  12. Done.

  13. In BE there is typically no comma after "i.e." and also in AE I don't think it's mandatory.

  14. Done.

  15. Done.

---

## Round 2 · Referee Report · Anonymous · 2018-4-19

Strengths
1) The paper is a pedagogical, self-contained review article, accessible to a non-expert audience.
2) The text is well-written and clear, with very few typos.
3) It contains easy exercises that keep the reader interested.
Weaknesses
None.
Report
I am satisfied with the changes made by the author.
Requested changes
No further changes required.

---

## Round 2 · Author Response

introduction added

---

## Round 2 · List of Changes

changes as stated in replies to referee reports

---

## Editorial Decision

published